# Sexual Health after a Breast Cancer Diagnosis: Addressing a Forgotten Aspect of Survivorship

**DOI:** 10.3390/jcm11226723

**Published:** 2022-11-14

**Authors:** Suneela Vegunta, Carol L. Kuhle, Jennifer A. Vencill, Pauline H. Lucas, Dawn M. Mussallem

**Affiliations:** 1Division of Women’s Health Internal Medicine, Mayo Clinic, 13400 Shea Blvd, Scottsdale, AZ 85259, USA; 2Menopause and Women’s Sexaul Health Clinic, Mayo Clinic, Rochester, MN 55905, USA; 3Department of Psychiatry and Psychology, Mayo Clinic, Rochester, MN 55905, USA; 4Department of Physical Medicine and Rehabilitation, Mayo Clinic, Scottsdale, AZ 85259, USA; 5Jacoby Center for Breast Health, Mayo Clinic, Jacksonville, FL 32224, USA

**Keywords:** preventative/mitigation strategies, risk factors and risk prediction, sexual dysfunction

## Abstract

Breast cancer is the most common cancer in women. The life expectancy after a breast cancer diagnosis is improving steadily, leaving many more persons with the long-term consequences of treatment. Sexual problems are a common concern for breast cancer survivors yet remain overlooked in both the clinical setting and the research literature. Factors that contribute to sexual health concerns in breast cancer survivors are biopsychosocial, as are the barriers to addressing and treating these health concerns. Sexual health needs and treatment may vary by anatomy and gender. Multidisciplinary management may comprise lifestyle modifications, medications, sexual health aids such as vibrators, counseling, and referrals to pelvic health physical therapy and specialty care. In this article, we review the contributing factors, screening, and management of sexual difficulties in cisgender female breast cancer survivors. More information is needed to better address the sexual health of breast cancer survivors whose sexual/gender identity differs from that of cisgender women.

## 1. Introduction

Breast cancer is the most common cancer in women [1,2]. The average life expectancy for breast cancer has improved dramatically, such that nearly 3.8 million women in the US are now alive after a diagnosis of invasive breast cancer [1]. Survivors of breast cancer represent more than 40% of all cancer survivors [3]. Breast cancer is inaccurately depicted as a “women’s disease” [4], but men and transgender and gender-diverse persons can also have breast cancer, although prevalence rates in these populations are more difficult to determine [5,6,7,8,9,10].

Cancer survivorship comes with its own set of challenges. Regardless of sex/gender or cancer type, most patients with cancer (hereafter called cancer patients) report cancer-related sexual concerns when asked, making sexual health recovery a prominent survivorship issue [11,12,13]. Sexual health is a vital sign for overall health and well-being, remaining relevant for most people throughout their lifespan. The World Health Organization defines *sexual health* as “a state of physical, emotional, mental, and social well-being in relationship to sexuality; it is not merely the absence of disease or infirmity.” Poor sexual health can affect emotional health, with decreased quality of life, depression, relationship problems, and low self-image, as well as physical health and social well-being [14].

Sexual health can be affected anytime during or after the course of breast cancer treatment. Approximately 50% of breast cancer survivors report some type of sexual difficulty, and 45% specifically report sexual pain [15,16]. Despite this reported frequency of sexual difficulties, adequate sexual health recovery remains an unmet need; numerous barriers exist for cancer survivors to access sexual health services. Health care professionals (HCPs) may lack the knowledge, time, prior training, and resources or may unnecessarily fear embarrassing patients by addressing their sexual health [11,17]. Demographic differences may also be a barrier to sexual health treatment, with breast cancer patients indicating that they are more likely to discuss sexual concerns with an HCP of a similar perceived age or sex/gender [18].

Most research on the intersection of sexual health and breast cancer has focused on cisgender women (persons assigned female sex at birth and identifying as women in gender). A paucity of information exists regarding the sexual health needs of cisgender men or transgender and gender-diverse breast cancer survivors and how breast cancer treatments affect their body image. Therefore, cisgender women will be the focus of this article. To provide help and guidance to HCPs on this topic, we aimed to review the contributing factors, screening, and management of sexual difficulties in cisgender female breast cancer survivors. Herein, we summarize the literature pertaining to sexual health after breast cancer treatment, evaluation, and management.

### Methods

We performed a literature search in PubMed, Google Scholar, and Scopus for peer-reviewed, English-language publications published from January 2010 to March 2022 using relevant search terms: anorgasmia, assessment, breast cancer survivor, cancer treatment, libido, sexual dysfunction, sexual function, sexual health, sexuality, and treatment. Additional search terms included flibanserin, bremelanotide, bupropion, vaginal estrogen, and vaginal DHEA, as well as various combinations of pelvic physical therapy AND dyspareunia AND breast cancer. We did not attempt to summarize studies or data from existing studies in a systematic way. Articles were chosen for inclusion based on judgment and interpretations of findings. A total of 900 articles were reviewed, including citations from prospective articles. Articles not pertinent to breast cancer were excluded.

## 2. Sexual Health of Breast Cancer Survivors

Sexual function is complex and is affected by biological, psychosocial, and sociocultural factors that may promote or inhibit sexual response [19]. Among the components of the sexual response cycle are desire, arousal, and orgasm. Sexual desire, or libido, is influenced in multifaceted ways: by the physical and psychological health of the patient, past sexual experiences, sex education, relationship concerns (if partnered), personal beliefs about sex, and many other factors. Arousal is a neurovascular response to desire or sexual stimulation resulting in vascular congestion of the nipples/areolae and genitals. Arousal may produce an orgasm, a marked feeling of sexual release followed by rhythmic contractions in the pelvic musculature. For breast cancer survivors, it is notable that nipple-areolar sensation can be an essential component of arousal and orgasm [20,21].

Assessing sexual difficulties includes evaluating the patient’s sexual response and pain. Sexual pain may occur with stimulation of the vulva and clitoris, with initial or deep vaginal penetration, or with both [22]. Sexual pain can also result from conditions such as genitourinary syndrome of menopause (GSM, previously termed vulvovaginal atrophy), pelvic floor dysfunction, or chronic pelvic pain syndrome, to name a few.

## 3. Contributing Factors

Factors that can affect the sexual health of breast cancer survivors include the following [23]: advanced age, preexisting sexual problems, and poor body image, in addition to the type of breast cancer treatment (i.e., multimodal treatments, chemotherapy, and long-term endocrine therapy). Sexual function can be affected by many other coexisting factors, such as underlying medical conditions, which may have a varying impact on sexual response [24,25]. For example, uncontrolled diabetes may affect pelvic engorgement and arousal because of changes in neurovascular function [26,27]. Cardiovascular disease may reduce sexual response through vascular compromise or the use of prescribed medications. Antidepressants and antihypertensive medications may diminish libido, arousal, and orgasm [28]. Menopausal status also has a role. Postmenopausal women may have worsening vulvovaginal dryness and discomfort with sexual activity. Premenopausal women, however, often have more aggressive tumor or cancerous cell types requiring more aggressive local and systemic breast cancer treatments. These treatments can result in more severe sexual dysfunction and body image–related difficulties than in postmenopausal patients undergoing breast cancer treatment [29,30].

## 4. Impact of Breast Cancer Treatments on Sexual Function

Treatment-related sexual adverse effects are one of the most prevalent and distressing aspects of cancer therapy. Surgical treatments, systemic therapies, and radiation therapy all can lead to sexual health problems. Patients who have received multimodal breast cancer treatments are at high risk for sexual function problems [31]. A summary of how breast cancer treatments can affect sexuality appears in Table 1.

### 4.1. Surgical Treatment

Alteration of breast anatomy by lumpectomy or mastectomy may negatively influence sexual self-image, self-esteem, and confidence, affecting sexual intimacy and overall sexual well-being [32,33,34]. Many surgery-related changes can be long-term and disfiguring, such as loss of breast volume, asymmetry, fat necrosis, and occasionally seromas, which can develop after lumpectomy. Removal of the breasts and nipples can not only be disfiguring but can also diminish breast-related sensation and arousal. Occasionally, nerve injury can cause hyperesthesia or dysesthesia of the chest wall, further disrupting potentially pleasurable sexual sensations [35].

The type of surgical modality seems to influence sexual health. Women who had breast lumpectomy reported significantly greater satisfaction with breast appearance and more pleasurable breast caress than women who had a mastectomy with reconstruction (*p* = 0.002 and *p* = 0.01, respectively) [36]. Aerts et al. [37] reviewed the type of breast surgery and its impact on sexual function and found that women who underwent breast-conserving surgery had better sexual adjustment and reported less impairment of sexual desire, arousal, and orgasm than women who underwent mastectomy. Nipple-sparing mastectomy has been associated with better psychosocial and sexual well-being than skin-sparing mastectomy followed by nipple reconstruction [38,39,40]. Breast reconstruction with nipple-areolar reconstruction is associated with improved results on the BREAST-Q psychosocial and sexual well-being scales (*p* < 0.002 and *p* < 0.001, respectively), and this improvement did not change over time [41]. Other surgical aspects in node-positive breast cancer, such as axillary lymph node removal, can lead to lymphedema of the chest wall, axilla, and arm [42] and can affect sexuality. In a small study by Mulkoglu et al. [43] comparing women who underwent mastectomy and who had similar emotional scores with and without lymphedema after breast cancer treatment, women with lymphedema had poorer sexual function and quality of life.

Some premenopausal patients with hormone receptor-positive breast cancer may require a prophylactic bilateral salpingo-oophorectomy and possibly long-term endocrine therapy to improve treatment outcomes. This treatment can lead to premature surgical menopause, which is a major risk factor for sexual difficulties [44].

### 4.2. Chemotherapy

Taxanes, platinum agents, and anthracyclines are chemotherapy agents routinely used in the treatment of breast cancer. Chemotherapy in young, premenopausal breast cancer patients can cause abrupt and often permanent premature menopause, with a substantial impact on sexual desire [45,46]. Two distinct end points of decline in sexual desire in women diagnosed with breast cancer have been identified: after diagnosis of breast cancer and after receiving the first cycle of chemotherapy [47]. Chemotherapy-induced ovarian insufficiency may lead to genital problems with loss of lubrication and elasticity of the vagina, resulting in dyspareunia and a decline in libido [15,48,49]. Fatigue, pain, and insomnia related to chemotherapy can also affect sexual drive. Some chemotherapeutic agents cause both central and peripheral neuropathy, leading to loss of sensation, abnormal genital sensation, tingling, numbness, and anorgasmia. Some chemotherapeutic agents affect motor and autonomic nerves that lead to incontinence of the bowel and bladder, which can leave women afraid to engage in sexual activity. Many chemotherapeutic agents cause alopecia, which can further worsen a patient’s physical and sexual body image.

### 4.3. Radiation Therapy

Radiation therapy for breast cancer can reduce locoregional and metastatic disease and can improve the survival rate, but it has associated acute and chronic toxicities. Acute toxic effects may include fatigue, esophagitis, breast edema, induration, dermatitis, and subsequent hyperpigmentation of the skin over the breast. Chronic toxic effects may include radiation pneumonitis, lymphedema, hypothyroidism, and cardiac toxicity [50]. In addition, radiation therapy can affect the aesthetic outcomes of reconstructive surgery [51]. All these toxic effects can negatively influence sexual function.

### 4.4. Adjuvant Endocrine Therapy

About one-third of women diagnosed with breast cancer are premenopausal, and overall, 75% of breast cancers are estrogen receptor-positive [2]. Hormone receptor-positive breast cancer treatment typically includes the use of endocrine therapy to prevent the recurrence of breast cancer and to improve overall survival [52]. Endocrine therapy includes selective estrogen receptor blockers (SERMs), such as tamoxifen, and aromatase inhibitors (AIs). Tamoxifen can be used as adjuvant endocrine therapy in both premenopausal and postmenopausal women. AIs block estrogen production, thus causing a hypoestrogenic state, and can be used for premenopausal patients receiving concomitant ovarian suppression with the gonadotropin-releasing hormone agonist leuprolide (Lupron), for premenopausal patients who undergo bilateral salpingo-oophorectomy, and for postmenopausal patients. Ovarian suppression results in the abrupt loss of ovarian function, leading to GSM with vaginal dryness and loss of elasticity; this condition often results in sexual pain, decreased arousal, loss of desire, and poor sexual satisfaction [53,54].

Unlike chemotherapy, which is for a short duration, adjuvant endocrine therapy is administered over extended periods, usually 5 to 10 years. SERMs can cause hot flashes, endometrial cancer, and polyps, and AIs can cause osteopenia, arthralgias, hot flashes, and fatigue [55,56]. SERMs and AIs can cause vulvovaginal changes leading to dyspareunia and low libido. In a study comparing these 2 classes of medications, women receiving tamoxifen reported sexual concerns less often than women receiving AIs (*p* = 0.05) [57]. Significantly more women receiving AIs reported vaginal dryness and dyspareunia [57]. In another prospective observational study, hypoactive sexual desire disorder (HSDD) was more prevalent in sexually active, current AI users (66.7%; 95% CI, 49.4–83.9) vs. current nonusers (43.6%; 95% CI, 37.0–50.2; *p* = 0.02) [58]. Women who have adverse effects from one form of endocrine therapy may be switched to a different drug in the same class or a different class to manage adverse effects as indicated.

### 4.5. Immune Therapy

Immune checkpoint blockade has been found to help patients with triple-negative breast cancer, resulting in longer event-free survival compared with chemotherapy alone [59]. This treatment modality is still in the early stages of development. Currently, the longer-term influence of immune therapy on breast cancer survivors and its influence on sexual health is not known and needs additional research. Immunotherapy has been used to treat other cancers and is known to cause substantial toxicities. Common adverse effects are colitis, pneumonitis, hepatitis, and nephritis. Immunotherapy can also lead to endocrinopathies such as diabetes mellitus, hypoparathyroidism, hypothyroidism, and even adrenal insufficiency. Practitioners who provide care to breast cancer survivors need to be able to recognize these adverse effects. In addition, these effects can influence sexual health due to the presence of fatigue, poor quality of life, and various endocrinopathies. This area needs further research.

## 5. Assessing Sexual Health in Breast Cancer Survivors

Challenges exist in discussing sexual health difficulties with breast cancer survivors. Most cancer care professionals are not well versed regarding the impact of cancer therapeutics on their patients’ sexual health and are not comfortable discussing this topic with patients [60,61]. Despite this, research findings consistently show that patients consider sexual health to be an important aspect of their health and prefer that their HCPs initiate conversations about sexual health difficulties and possible treatment options. In a US poll of 500 adults, 71% reported that they believed their physician might dismiss sexual concerns if they raised the topic [62]. Similarly, less than 50% of oncologists surveyed in another study reported that they routinely asked questions to assess patients’ sexual health concerns, a finding suggesting that communication about sexual health is a major challenge in the oncology setting [63].

Of concern is that, left untreated, sexual problems are associated with reduced quality of life, depression, interpersonal conflicts, and often medication noncompliance [64]. Early diagnosis of sexual difficulties followed by intervention leads to better sexual health outcomes and improved quality of life. This evidence makes it critical for HCPs to inform patients about safe and effective treatments for sexual difficulties. HCPs can start conversations about sexual health by asking their breast cancer patients questions such as those in Box 1.

Box 1Examples of How to Ask a Patient About Sexual Health Issues.“Are you currently sexually active?” If not, “Does this concern you?”“Many women have sexual concerns during cancer treatment. Do you have any concerns you would like to discuss today?”“Have you noticed any change in your sexual interest?” If so, “Does this concern you?”“Are you having any difficulty with vaginal lubrication or pain with sexual intercourse?”“Do you have concerns about your sexual well-being?”

The American Cancer Society and the American Society of Clinical Oncology, in their joint Breast Cancer Survivorship Care Guideline [65], and the National Comprehensive Cancer Network [66] recommend assessing breast cancer survivors’ body image and sexual function at regular intervals. The Brief Sexual Symptom Checklist for Women [67] is a simple tool for use regardless of sexual orientation, the current level of sexual activity, or partner status [65,66,68] (Table 2). Several validated questionnaires exist for evaluating female sexual health. Administering the questionnaires before and after a woman starts breast cancer treatment can help assess sexual health concerns. Some commonly used questionnaires are the Female Sexual Function Index (FSFI), the Sexual Activity Questionnaire, and the Decreased Sexual Desire Screener. The FSFI is widely used as a measurement of female sexual function in the field of sexual medicine and for cancer survivors and is considered the criterion standard. It is a 19-item, patient-reported outcome measure to evaluate female sexual function in 6 separate domains: sexual desire, arousal, lubrication, orgasm, satisfaction, and pain. A woman with a total FSFI score of less than 26.55 is diagnosed with female sexual dysfunction.

## 6. A Multidisciplinary Approach for Improving Sexual Health after Breast Cancer Treatment

Female sexual health is complex, and managing sexual difficulties in women needs a multidisciplinary team approach. Because sexual problems in breast cancer survivors are multifactorial, the initial step is to promptly identify the contributing factors for sexual problems. Then, the treatment approach includes the management of vaginal dryness and sexual pain with vaginal lubricants, massagers, moisturizers, local hormone therapy when not contraindicated, and referral to a pelvic floor physical therapist. Management of mood disorders, sexual self-image concerns, and relationship issues may need a referral to a sex therapist. Many patients will see substantial improvement with these measures, but patients who need additional intervention can be considered for medication management. Medications should be individualized based on the patient’s clinical presentation. For example, patients with depression can be treated with bupropion, which has minimal sexual adverse effects.

### 6.1. Impact of Lifestyle on Sexual Function

Poor physical health is a major risk factor for female sexual dysfunction. Conditions such as metabolic syndrome [69], diabetes [70], hypertension [71], dyslipidemia, and obesity [72] predict female sexual difficulties [73,74,75]. Adhering to the healthy lifestyle recommendations set forth by the American Cancer Society and the American Institute for Cancer Research (AICR) was associated with reduced disease-specific and overall mortality and improved health-related quality of life (HRQoL) among breast cancer survivors [76,77]. These recommendations included managing one’s weight; following a whole food, plant-predominant dietary pattern; regularly exercising; and avoiding alcohol and tobacco use. These same lifestyle elements are linked to improved sexual function.

A tailored health education program centered on the AICR recommendations was part of a 12-week healthy living program that also provided clinical consultations; 55 breast cancer survivors aged 45 to 60 years were randomly assigned to either the intervention group or a control group [78]. At 12 weeks, women in the intervention group reported clinically significant reductions in sexual dysfunction, vasomotor symptoms, somatic symptoms, and overall menopausal symptoms compared with the control group.

The direct impact of nutrition on sexual function among women is not yet fully elucidated; however, nutritional improvements in body weight, cardiometabolic health, vascular endothelial function, blood glucose regulation, and mood may have a role in improving sexual function [79]. Investigators of multiple randomized controlled trials and cross-sectional studies have examined the effect of the Mediterranean diet on reported sexual function and found that adherence to this eating plan resulted in better sexual function in a dose-dependent manner [80]. More than half of all women who undergo adjuvant chemotherapy for early-stage breast cancer have substantial gains in fat mass and reduction in lean mass [81]. Only 10% of women with breast cancer-related weight gain return to their pretreatment weight [82]. Weight gain after breast cancer treatment is associated with poorer breast cancer-related morbidity and mortality, and body image changes are related to high psychological distress. The AICR Continuous Update Project reported strong evidence linking the Mediterranean dietary pattern to a decreased risk of being obese or overweight and to reduced weight gain. Weight management efforts are known to improve breast cancer outcomes [83]. Achieving ideal body weight is known to restore physical health, which could improve sexual function. Additionally, a randomized study of 38 postmenopausal women found that a low-fat, vegan diet supplemented with 86 g (0.5 c) of whole soybeans daily improved quality of life in vasomotor, sexuality, physical, and psychological domains [84].

Regular exercise has been shown in some studies to prevent the development of sexual dysfunction in middle-aged women and to maintain breast cancer survivors’ sexual activity. In a prospective study, 537 breast cancer survivors aged 35 to 68 years were randomly assigned to a 12-month exercise trial and followed up for 5 years; their HRQoL was compared with that of the age-matched, female general population. The mean HRQoL was statistically and clinically impaired for breast cancer survivors vs. the general population, although the overall mean HRQoL of breast cancer survivors did not improve significantly over 5 years; however, the domains of sexual activity, usual activities, depression, and distress did improve from baseline [85,86].

Researchers have reported the effects of tobacco and alcohol use on sexual function in the general population. Tobacco use has been linked to a 30% reduction in genital arousal, delayed orgasm, and reduced vaginal lubrication, and those with alcohol dependence had sexual problems and reduced vaginal lubrication [78,87].

Causes of sexual problems among breast cancer survivors are multifactorial, and a lifestyle modification approach to care should be considered. A diagnosis of cancer has been associated with favorable behavior change, enhanced self-care, self-control, and optimism such that a breast cancer diagnosis may be a teachable moment for achieving adherence to healthy living [88]. The need exists to prioritize and fund high-quality research regarding the impact on breast cancer survivors’ sexual health of following an overall pattern of healthy living, including a plant-predominant diet, physical activity, and avoidance of toxic substances.

### 6.2. Managing GSM

*Genitourinary syndrome of menopause* is a broad term and includes physical symptoms such as vulvovaginal symptoms, dyspareunia, and urinary tract symptoms. Symptoms of GSM can be managed effectively for breast cancer patients and survivors. Treatment options are guided by stage and hormone receptor status; however, controversy exists among clinicians regarding the safety of vaginal estrogen therapy, vaginal dehydroepiandrosterone, and ospemifene, especially for patients with a history of estrogen receptor-positive breast cancer. The paucity of evidence regarding the safety of vaginal hormone therapies for breast cancer patients has led to some patients avoiding treatment of GSM, which potentially decreases their quality of life and hurts their intimate relationships [89]. Factors that influence decision-making regarding GSM treatment include the severity of symptoms, risk of breast cancer recurrence, response to previous treatments, and personal preferences [89]. Shared decision-making, which centers on these factors, and informed consent are required when HCPs recommend treatment of GSM to breast cancer patients [89]. Available treatments of GSM as related to breast cancer survivors are shown in Table 3 and Table 4 [90,91,92,93,94,95,96,97,98].

### 6.3. Medication Options for Managing Low Sexual Interest and Arousal

To date, two medications are approved by the US Food and Drug Administration (FDA) for the treatment of HSDD in premenopausal women: flibanserin and bremelanotide. These medications do not have a hormonal mechanism for improving sexual function, but they have not been well studied in breast cancer survivors, and additional research is needed in this population. The antidepressant bupropion, which is not FDA approved for HSDD treatment, is an option for breast cancer survivors with depression and associated sexual concerns.

#### 6.3.1. Flibanserin

Flibanserin works as a 5-HT_1A_ receptor agonist and a 5-HT_2A_ receptor antagonist; it is FDA-approved as a daily oral therapy for the indication of acquired, generalized HSDD in premenopausal women in the US [99]. In Canada, flibanserin also is approved for postmenopausal women aged 60 years or younger [100]. The drug’s exact mechanism of action is unknown. Results of several studies showed that flibanserin has a positive influence in both premenopausal and postmenopausal women on sexual desire, number of satisfying sexual events, and decreased sexual distress with clinically meaningful benefits [99,100,101,102,103,104]. Flibanserin, at a dosage of 100 mg daily, is generally well tolerated, except for central nervous system and gastrointestinal adverse effects [105]. Strong caution with alcohol intake is advised, as is attention to drug interactions [105]. In a small study of women with breast cancer receiving tamoxifen therapy, flibanserin over 24 weeks improved overall sexual function in most women [106]. A larger randomized placebo-controlled study is still needed.

#### 6.3.2. Bremelanotide

Bremelanotide is a melanocortin receptor agonist, which is a self-administered, on-demand subcutaneous therapy (1.75 mg) injected at least 45 min before sexual activity. It is FDA approved in the US for the treatment of premenopausal women with acquired, generalized HSDD [107,108]. Results of the RECONNECT Exit trial showed statistically and clinically significant enhancements in sexual desire and reduction of related distress with bremelanotide vs. placebo in women with HSDD who are premenopausal [109,110]. Other study findings showed improved desire, arousal, and orgasm scores when bremelanotide (1.75 mg) was given before sexual activity vs. a placebo [103,108,111,112]. The most common adverse effects were nausea (39.9%), facial flushing (20.4%), and headache (11.0%) [108]. Currently, information is lacking regarding bremelanotide specific to breast cancer survivors, but there is no contraindication against use in this patient population.

#### 6.3.3. Bupropion

Bupropion is a selective norepinephrine receptor inhibitor and is an antidepressant with a dual effect on norepinephrine and dopamine neurotransmitter systems with no known serotonergic activity [113]. Studies have noted that bupropion has the advantage among antidepressant medications of having less effect on sexual function and may even enhance sexual function for certain persons. Bupropion can be used as a first-line treatment option for breast cancer survivors with depression and associated sexual concerns.

### 6.4. Managing Other Patient Concerns Influencing Sexual Health

#### 6.4.1. Relationship Concerns

Cancer-related distress is a barrier to sexual health recovery and affects a breast cancer survivor’s coping skills in dealing with a breast cancer diagnosis and later with the impact of cancer treatment [114,115]. Breast cancer survivors have been shown to have notable anxiety, depression, and posttraumatic stress symptoms, which can persist well after treatment [116] and can directly or indirectly affect sexual health.

General distress and sexual distress related to breast cancer appear to vary by demographic and clinical factors. In a study of cisgender female breast cancer survivors and their cisgender male partners, older age was associated with less sexual distress [117,118]. Mental health concerns, such as depressive symptoms, were highest in survivors struggling with sexual arousal. Demonstrating the critical importance of partner and relational variables, Hummel and colleagues [118] documented that nearly two-thirds of the breast cancer survivors’ male partners had difficulties with erectile function. Partner erectile function was negatively associated with sexual pain affecting the breast cancer survivors, a common relational dynamic for many couples dealing with dyspareunia [119].

Sexual concerns often persist for years after patients complete breast cancer treatment. Partner and relationship factors, ranging from communication skills, level of sexual education, attachment and conflict management style, and partner sexual function, can exacerbate or buffer sexual health concerns for breast cancer survivors [120,121,122,123,124]. Additionally, recovery of sexual function and satisfaction is often predicated on patients’ ability to discuss sex with their current or potential partners, their attitudes toward sex, and any past sexual trauma for both the patient and the partner or partners [125]. HCPs should encourage communication between the patient and the patient’s partner regarding sexual concerns and recommend that they explore improving their sexual health, such as by using sexual health aids (e.g., vaginal moisturizers, vibrators, and lubricants) [11].

#### 6.4.2. Body Image

Substantial research documents a negative association between the impact of breast cancer treatment on body image and, in turn, sexual health for cisgender women [120,126,127,128]. In qualitative studies, such women have described their physical appearance after breast cancer treatment as “bizarre,” “weird,” or “does not look right” [129]. Body image concerns in breast cancer survivorship generally relate to two issues: (1) chest size and shape and (2) posttreatment scarring due to surgery or port placement for chemotherapy. Survivors also report body image and sexual health-related concerns around hair loss and weight gain [30,53,129]. Patients who had more than 20% of their breast tissue removed were reported to be at increased risk for dissatisfaction with postoperative scarring [130]. Because of their breast cancer surgical scars, cisgender women have reported discomfort being seen naked and often avoid nudity or even certain types of clothing to manage the concern [127].

Body image and sexual concerns related to cancer survivorship are common across all ethnic groups; however, important cultural differences exist. Results of a qualitative study of breast cancer survivors showed that African American, Asian American, and Latina women were more likely to receive mastectomies, and African Americans were less likely to receive adjuvant therapies [131]. African American patients with early-stage breast cancer reported a higher burden of symptoms than White patients. Future studies focusing on the impact of treatment-based outcomes among diverse racial/ethnic groups would help define the differences and diminish the disparities [29,132].

Altered body image can impact sexual function, and it requires a multimodal approach to improve sexual function [133]. A text-based online group therapy intervention involving 60 breast or gynecologic cancer survivors showed a significant reduction in distress about body image. Psychosexual dysfunction and quality of life improved but not significantly [134].

Sexual dysfunction can be especially difficult among young breast cancer survivors requiring long-term ovarian suppression. Results of a small study of young breast cancer survivors showed significant improvements in sexual function with a psychosexual group intervention that included health rehabilitation, body awareness exercises, and mindfulness-based cognitive therapy [53].

Breast cancer survivors can have sexual health difficulties because of changes to their breast anatomy and physiologic function. They can also have psychological and relational challenges compounded by their lack of communication about the effect of cancer treatments on sexual health [60,135]. Providing sexual health counseling to patients and their partners is critical, which ideally should begin before any treatment interventions and continue until their symptoms resolve [11,135,136,137]. The American Cancer Society/American Society of Clinical Oncology [65] and the National Comprehensive Cancer Network [66] recommend that HCPs promptly refer affected patients for multidisciplinary sexual health treatment. Referrals may be appropriate for any of the following: individual psychotherapy, couples counseling, gynecologic care, urologic care, physical therapy of the pelvic floor, and sexual health specialty care [65,66]. Access to such services, however, may be limited or unavailable in certain geographic areas.

### 6.5. Pelvic Health Physical Therapy

Pelvic health physical therapy (PHPT), also called pelvic floor physical therapy, is a specialty area within physical therapy focusing on pelvic and abdominal health problems [138]. This includes rehabilitation of the pelvic floor. Physical therapy frequently is considered the first-choice treatment of pelvic, abdominal, and low back dysfunction for breast cancer survivors because it is noninvasive, can be combined with other treatments, often results in symptom relief and improved function, has a low risk of adverse effects, and is cost-effective [139].

Pelvic health physical therapists provide evidence-based physical therapy services to address various disorders of the abdominal and lumbopelvic region, such as pelvic pain, diastasis rectus abdominis, vulvodynia, constipation, fecal or urinary incontinence, pelvic organ prolapse, and sexual dysfunction (e.g., vaginismus and dyspareunia). Licensed physical therapists who have undergone specialized pelvic health training address these difficulties in patients across the gender and sex spectra, although PHPT historically has been associated with women’s health, including breast cancer survivors, for its strong focus on pelvic trauma after childbirth. The pelvic physical therapist is trained to perform both external and internal pelvic floor examinations and uses interventions such as patient education, pelvic floor muscle (PFM) training with biofeedback techniques, bladder and bowel training, manual therapy, training with vaginal cones, and electrical stimulation.

#### 6.5.1. History and Examination

Management of sexual function for breast cancer survivors by a pelvic health physical therapist requires a detailed patient history to identify possible contributing factors and to direct the physical examination and treatment. The history-taking should include previous sexual function, current sexual status and limiting factors, and patient-reported symptoms of bowel and bladder dysfunction in addition to pelvic musculoskeletal symptoms and function. The physical examination addresses the neuromusculoskeletal structures from the rib cage and lower thoracic spine through the hips and thighs, and it typically includes a detailed evaluation of mobility and integrity of vulvar and pelvic muscles, fascia, and connective tissue. Many neuromusculoskeletal structures beyond the PFM can affect PFM function [140].

#### 6.5.2. Treatment Approach

Because sexual dysfunction in this patient population results from a combination of factors, a multimodal and individualized approach is required for pelvic physical therapy. Pelvic health physical therapists acknowledge that sexual dysfunction is a complex and global issue involving more than vulvar and vaginal tissues. The therapist carefully assesses all possible contributors to the patient’s sexual pain and dysfunction from the beginning of treatment to support realistic and positive treatment expectations [141].

Pelvic health physical therapists are trained to be sensitive to both physical and psychosocial aspects of patients’ sexual function and to talk with patients about topics such as body image, quality of intimate relationships, and their thoughts and beliefs about their condition or conditions. It is important to address both the physical and psychosexual mechanisms of dyspareunia through a multimodal treatment approach [142]. Concerns beyond the therapist’s scope of practice should prompt a referral to other practitioners. A multimodal 12-week physical therapy intervention for women with dyspareunia after gynecologic cancer treatment showed significant improvement in sexual distress, pain-related anxiety and catastrophizing, pain-related self-efficacy, depressive symptoms, and body image concerns. The PHPT intervention comprised education, PFM exercises with biofeedback, manual therapy, and home exercises [143]. Typically, PHPT combines weekly in-clinic sessions with a home program between sessions.

#### 6.5.3. Patient Education

New patients who have completed their breast cancer treatments and who experience pelvic floor function problems often have a limited understanding of what causes their sexual changes [144]. Education plays an important role and is a main component of PHPT for sexual dysfunction [145]. Topics include the anatomy of the pelvic floor and vulvovaginal area and the pathophysiology of dyspareunia. A mirror may be given to the patient during the pelvic examination to raise her awareness of her vulvar anatomy. Additional educational topics include pain-related biology as it relates to painful sexual intercourse, management of persistent pain, relaxation techniques, and the female sex response, as well as lifestyle modifications such as stress management, mindfulness, sleep hygiene, proper bowel and bladder habits; use of lubricants and moisturizers; and alternate ways to share intimacy besides vaginal penetration.

A randomized pilot trial compared 3 interventions for postmenopausal women diagnosed with early-stage breast cancer starting treatment with AIs [146]. The usual-care group received a handout on managing sexual and other adverse effects. The active treatment groups received a 6-month supply of vaginal moisturizer (with either hyaluronic acid or a prebiotic), a vaginal lubricant, a dilator, and access to a website and coaching. The authors concluded that sexual counseling helped the patients receiving AIs maintain stable sexual function. The treatment group had better outcomes at 6 months than the usual-care group, with those receiving the hyaluronic acid-based vaginal moisturizer having better improvement than those who received the prebiotic-based moisturizer [146].

#### 6.5.4. PFM Training

PFM training refers to exercises taught by HCPs, such as pelvic floor physical therapists, to improve the relaxation, coordination, strength, and endurance of the PFMs. These exercises were shown to have a positive effect on stress urinary incontinence and the symptoms of GSM, including vaginal dryness and dyspareunia. A study of postmenopausal women with GSM and urinary incontinence participating in a 12-week PFM therapy program found improved blood flow in vulvovaginal tissues, ability to relax the PFMs, and elasticity of vulvovaginal tissues [147].

Women experiencing dyspareunia after breast cancer treatment typically have myofascial pelvic pain and weakness, increased tone, and reduced length of the PFMs. Contributing factors include a hypoestrogenic state; tissue changes related to a hiatus in vaginal penetration and sexual intimacy for the duration of the cancer treatments; anxiety, depression, or both; changes in body image; pain; fatigue; and gastrointestinal distress. The goals of PFM therapy are improving the circulation, relaxation, and lengthening of the tissues around the vagina and improving the awareness and coordination of the PFMs. Often, PFM therapy is enhanced by digital feedback (patient or therapist), visual feedback (use of a mirror), real-time ultrasonography, or the use of pressure or surface electromyography-type equipment.

#### 6.5.5. Manual Therapy

Manual techniques such as myofascial release can be directed to the vulvar area as well as intravaginally or intrarectally. The aim of these techniques is to increase the woman’s awareness of her PFMs, promote genital blood flow, improve tissue flexibility, and reduce muscle tension and musculoskeletal pain [145].

Results of several studies on manual therapy for the treatment of dyspareunia in women showed substantial improvements in the pain domain of the FSFI [148]. The patients were instructed to perform external and internal pelvic floor self-massage between physical therapy sessions and after physical therapy concluded. This can be accomplished either manually or with the use of special pelvic floor massage wands or vibrators.

#### 6.5.6. Insertion Exercises and Dilator Training

Vaginal insertion by a finger or dilator often is incorporated into the treatment of women who have pain with penetration. These exercises are meant to stretch the vaginal tissues and release tension in them by gradually increasing the size of the dilator inserted [149]. Because the aim of insertion practice is to desensitize the tissues, it serves as a graded exposure strategy. Through the regular practice of mindfully inserting the dilator while practicing relaxation techniques—both general and specific for the PFM—patients can gradually reduce their fear of painful penetration and improve their confidence about resuming sexual activity with penetration. Generally, clinicians recommend that patients use the dilator without their partner, so they are in full control of the exercise, before attempting partnered penetration. Many clinicians recommend the use of insertion techniques three times weekly, but optimal parameters for dilator use have not been established for preserving or improving vaginal size, either during cancer treatment or after menopause. Meditating, playing relaxing music, and using the dilator for more than 3 months are factors that may result in better outcomes [150]. Limited adherence to dilator use has been observed in multiple studies [151]. The pelvic health physical therapist can provide accountability and help support patient treatment adherence during follow-up sessions by using motivational interviewing and other coaching techniques [152].

## 7. Sexual Health in Marginalized Patient Populations

Patients with often-marginalized sexual orientations and gender identities, e.g., lesbian, gay, bisexual, transgender, and queer (LGBTQ), are more likely to report distress related to their cancer diagnosis and less likely to access existing resources that are geared toward heterosexual and cisgender patients [153]. In addition, LGBTQ-specific sexual health concerns are often left unaddressed by practitioners [154,155], and misinformation or a lack of information specific to patients with these marginalized identities remains common. To address these identified gaps in care, the World Health Organization and LGBTQ-serving organizations developed guidelines supporting the creation of intake forms that collect data on sexual orientation and gender identity and strongly encourage LGBTQ-specific cultural competency training [156,157]. More information is needed to better address the sexual health of breast cancer survivors whose sexual and/or gender identity differs from that of cisgender, heterosexual women.

## 8. Conclusions

Breast cancer survivors should be considered at high risk for sexual difficulties because preexisting health conditions may worsen, and new sexual health problems often develop. All breast cancer treatment modalities can lead to sexual health problems through various mechanisms. Patients should be queried early and frequently by their health care team regarding their sexual health. Sexual health is complex, and sexual problems are multifactorial and should be addressed thoroughly with survivors. Multiple treatment options exist for sexual problems, and these options should be discussed with the patient. A multidisciplinary approach usually gives the best results in the treatment of breast cancer patients with sexual dysfunction.

## Figures and Tables

**Table 1 jcm-11-06723-t001:** Mechanisms Through Which Breast Cancer Treatments Can Influence Sexual Function.

Type of Treatment	Mechanism of Sexual Dysfunction
Cancer surgery	Body image concerns Loss of sensation of nipple, breast, and/or chest Lymphedema
Chemotherapy	Premature menopause GSM with sexual pain Alopecia leading to body image concerns Anxiety related to cancer diagnosis Treatment-related weight gain, fatigue, and neuropathy Pelvic floor problems
Radiation treatment	Painful dermatitis Early/premature menopause GSM Loss of nipple sensation
Endocrine therapy	Premature/early menopause GSM Sexual pain due to hormonal insufficiency Treatment-related myalgias, fatigue, weight gain, and vasomotor symptoms with sleep disruption

Abbreviation: GSM, genitourinary syndrome of menopause.

**Table 2 jcm-11-06723-t002:** Brief Sexual Symptom Checklist.

1. Are you satisfied with your sexual function?
2. How long have you been dissatisfied with your sexual function?
3. The problem(s) with your sexual function is: Little or no interest in sex Decreased genital sensation (feeling) Decreased vaginal lubrication (dryness) Reaching orgasm Pain Other
4. Which problem is most bothersome?
5. Would you like to talk about it with your doctor?
National Cancer Care Network (NCCN.org) version 2.2014

**Table 3 jcm-11-06723-t003:** Nonpharmacologic and Nonhormonal Treatment Strategies for GSM Not Contraindicated for Use by BC Patients.

Treatment	Product Details	Use	Effects, Evidence
Vaginal lubricant	First-line therapy, low risk, nonhormonal; water-, silicone-, or oil-based	As needed for sexual activity to increase comfort and pleasure; can be used with other therapies	Does not reverse cellular/pH changes of GSM
Vaginal moisturizer	First-line therapy, low risk, nonhormonal	Daily to maintain tissue integrity and elasticity; can be used with other therapies	Does not reverse cellular/pH changes of GSM; a randomized clinical trial was performed to compare a 10-μg vaginal estradiol tablet vs. placebo gel vs. placebo tablet plus vaginal moisturizer vs. dual placebo, but neither the vaginal estrogen nor the moisturizer was superior to the placebo [90]
Hyaluronic acid gel	Low risk, nonhormonal, hydrating properties	Once every 3 days for a total of 10 applications over 30 d	Reduced clinical symptoms of vaginal dryness in a randomized controlled trial comparing hyaluronic acid vaginal gel with estriol cream [98]
Vibrator	Stimulates vulvovaginal tissues and natural lubrication	Alone or with partner	Reduced pain with vaginal penetration in a pilot study among patients without breast cancer [90]
Dilator	Stretches vaginal tissues	Ideal duration and frequency of use are unknown	NA
Fractional CO_2_ laser	Not FDA-approved for GSM	Generally delivered as 3 treatments 1 month apart	Limited data available had mixed results; some study findings suggested laser intervention can induce morphologic vaginal changes, alleviate vaginal dryness and dyspareunia, and improve sexual function, sexual distress, and the Wong–Baker Faces scale pain rating, whereas other findings showed no significant improvement over sham after 12 months [91,92,93]; safety is not yet proved by large-scale, randomized placebo-controlled trials

Abbreviations: BC, breast cancer; CO_2_, carbon dioxide; FDA, US Food and Drug Administration; GSM, genitourinary syndrome of menopause; NA, not available.

**Table 4 jcm-11-06723-t004:** Pharmacologic Treatment Strategies for GSM.

Medication	Dosage, Route	FDA Approval	Effects, Evidence	Other Considerations	No Contraindication for BC Patients
Topical lidocaine	4% aqueous lidocaine, vaginal introitus a few minutes before sexual activity	Yes	Does not improve cellular or pH changes of GSM; significantly improved worst pain score from 5 to 0 vs. normal saline, which provided minimal relief (from 6 to 4) in an RCT of 49 postmenopausal BC survivors	Consider in combination with other therapies (e.g., lubricants, moisturizers, vibrators, dilators, pelvic floor PT)	No contraindication for BC patients
DHEA (prasterone)	6.5 mg/d, vaginally	Yes for treatment of moderate to severe dyspareunia, but the label warns against use in BC patients, who were not included	In an RCT of cancer survivors (mostly BC survivors) who received either compounded 3.25-mg or 6.5-mg daily doses of DHEA or placebo, the estradiol and testosterone levels that were within the lower half of values in the postmenopausal range significantly increased for 6.5-mg/d dose but not 3.25-mg/d dose in the same study, sexual function improved significantly with 6.5-mg dose vs. placebo, but vaginal dryness and dyspareunia did not improve with either dose [95,96,97]	NA	No contraindication for BC patients, but limited studies with BC survivors exist; should be prescribed after discussion with oncologist
Local estrogen [94]	17 β-Estradiol as 10-μg tablets inserted intravaginally daily for first 14 d, then twice weekly; or as a vaginal ring, placed intravaginally every 90 d	No for patients with BC	Absorption varies depending on potency, type, and formula; where the product is applied; and whether skin is atrophied or estrogenized. Lacks evidence to support increased risk of BC recurrence with higher estrogen levels within a narrow postmenopausal range. In a small study of BC survivors, twice-weekly vaginal estrogen therapy improved sexual function and vaginal health better than nonhormonal vaginal moisturizers; no significant effect was found on endometrial thickness or systemic estrogen levels.	Systemic estrogen levels should not be used in clinical decision-making to monitor local estrogen therapy. Treatment requires discussion of risks and benefits and review with the patient’s oncologist. Factors affecting decision-making include BC stage, grade, lymph node involvement, hormone receptor status, use of endocrine therapy, risk of recurrence, time since diagnosis, symptom severity, nonhormonal options available, and impact on quality of life.	No contraindication for BC patients and endorsed by ACOG and ACS for use by BC survivors after discussion with oncologist

Abbreviations: ACOG, American College of Obstetricians and Gynecologists; ACS, American Cancer Society; BC, breast cancer; DHEA, dehydroepiandrosterone; FDA, US Food and Drug Administration; GSM, genitourinary syndrome of menopause; NA, not available; PT, physical therapy; RCT, randomized clinical trial.

## Data Availability

Not applicable.

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
