# Peer review of "Sexual Health after a Breast Cancer Diagnosis: Addressing a Forgotten Aspect of Survivorship"

_jcm, 2022, doi:10.3390/jcm11226723_

Round 1

Reviewer 1 Report

Dear authors, 

thank you for submitting your draft 'Sexual Health after a breast cancer diagnosis: addressing a forgotten aspect of survivorship' for review. 

As clinician I agree 110% with your statement that this is a very often forgotten aspect of survivorship. Looking forward to your review. There were several clinical aspects that are interesting to read. As you point out the sexual health is a very complex system which can easily be disturbed by a life threatening disease your review could provide further in depth information for clinicians. Like which screening tool (FSFI, FSFI-BC, EORTC, HRQoL ...) to use or which communication strategy might be most promising (PMID: 31762399 and others). From my experience patients with the need for prophylactic bilateral salpingooophorectomy are most likely BRCA mutated. I.e. have a completely different view on life and children compared to 'normal' BC patients. To state that the possible premature menopause (caused by the surgery) is a major risk factor for sexual difficulties in BRCA+ patients should be backed by a good reference. Especially in such a complex topic like female sex health. Similar the study of Mülkoglu et al. 25 patients +/- lymphedema without an information on tumor stage, removed lymph nodes and further treatments again is from my viewpoint a bit short fetched and should provide more detailed information on which part of the FSFI is most likely impaired. 

Further the section on chemotherapy is missing the current treatment options like antibodies. These long term treatments may impair the 'normal' post chemotherapy life. How do BC patients compare to other chronic illnesses like dialysis or high blood pressure patients?

You provide several medical and non medical treatment options - some of which are clearly contraindicated in certain breast cancers in the early follow up time or not approved by the FDA. It would be desirable to have an algorithm/flow chart that would easily 'guide' the reader/care provider through these options. In the current version the treatment options are very general and non specific for BC patients.

I would recommend rewriting the article focussing on BC patients and a validated screening tool for sexual health and providing the possible treatment options with current literature and if no BC data are available use most likely data instead. 

Also some information on the literature search/method would be appreciated as the search term 'sexual health breast cancer' in pubmed provided some interesting not quoted publications of recent years. 

As mentioned earlier I do know that there is a strong demand for such a review in the clinical practice. So please understand my comments as constructive indications and make the best out of them. 

Author Response

[Y]our review could provide further in depth information for clinicians. Like which screening tool (FSFI, FSFI-BC, EORTC, HRQoL ...) to use or which communication strategy might be most promising.

Authors’ response: Thank you for this valuable suggestion. Screening is an important aspect of evaluating a breast cancer survivor for sexual difficulties, and we have added an additional paragraph detailing sexual assessment questionnaires in section 5 on pages 7-8.

Similar the study of Mülkoglu et al. 25 patients +/- lymphedema without an information on tumor stage, removed lymph nodes and further treatments again is from my viewpoint a bit short fetched and should provide more detailed information.

Authors’ response: Thank you for this suggestion. We added more details on page 5 regarding this study and an additional reference (Gillespie et al) to validate this point.

Further the section on chemotherapy is missing the current treatment options like antibodies. These long-term treatments may impair the 'normal' post chemotherapy life. How do BC patients compare to other chronic illnesses like dialysis or high blood pressure patients?

Authors’ response: Immune therapy for breast cancer is a promising advancement in selected breast cancer patients. Breast cancer survivors who received immune therapy and the longer-term sequelae of this treatment are not well studied, making immune therapy an area of opportunity for additional research. We added a separate section (4.5 on page 7) on immune therapy. Regarding the reviewer’s second point, currently, no known comparative studies exist regarding sexual health in women with breast cancer and women with chronic diseases.

You provide several medical and non-medical treatment options - some of which are clearly contraindicated in certain breast cancers in the early follow up time or not approved by the FDA. It would be desirable to have an algorithm/flow chart that would easily 'guide' the reader/care provider through these options. In the current version the treatment options are very general and non-specific for BC patients.

Authors’ response: Most of the treatment options that we discussed are used for breast cancer survivors without any adverse effects and with subsequent improvement. Unfortunately, many effective treatments have not yet received FDA approval, and they are used off-label by many practitioners. Some medications such as flibanserin have limited data, and bremelanotide has no data from breast cancer survivors; we added this information to the manuscript (pages 10 and 11). No medication listed in Table 2 is contraindicated for use by breast cancer patients, and this information was added to the table title. Table 3 lists the products to treat vaginal dryness, and only ospemifene is contraindicated for breast cancer survivors; we have deleted this medication from Table 3 and added an additional column to validate use of the other medications for breast cancer survivors. Additionally, on page 9, we added “as related to breast cancer survivors” in the last sentence of section 6.2, referring to Tables 2 and 3.

would recommend rewriting the article focussing on BC patients and a validated screening tool for sexual health and providing the possible treatment options with current literature and if no BC data are available use most likely data instead. 

Authors’ response: We thank you for this suggestion. We have made major changes to our manuscript by rewriting portions of the manuscript (including adding several mentions of “breast cancer survivors” on page 13), adding additional paragraphs, and deleting information that is not relevant to breast cancer survivors, adding several additional references, and modifying our table. We believe that your valuable suggestions have helped improve our manuscript.

[S]ome information on the literature search/method would be appreciated as the search term 'sexual health breast cancer' in pubmed provided some interesting not quoted publications of recent years.

Authors’ response: We agree with this suggestion and have added this additional information on our search parameters as recommended (Introduction on page 3).

Reviewer 2 Report

Dear Authors,

Well done. This is a well-written review paper. As authors mentioned, breast cancer survivors should be considered at high risk for sexual difficulties because preexisting health conditions may worsen and new sexual health problems often develop. But, most doctors are only interested in treatment of breast cancer.

1.     I suggest performing a careful revision to correct some typos and revise grammar.

2.     In introduction;

A.     When referring to epidemiology on breast cancer, try to use the latest (2022) data as much as possible.

B.     Authors said that the subject of this article is focused on cis-gender women. If so, I recommend to briefly reduce the cisgender men, transgender and gender-diverse contents mentioned in the introduction.

3.     In 4.1. Surgical Treatment

A.     Add a reference to the contents “Occasionally, nerve injury can cause hyperesthesia or dysesthesia of the chest wall, further disrupting potentially pleasurable sexual sensations.”

B.     Add a reference to the contents “Some premenopausal patients with hormone receptor–positive breast cancer may require a prophylactic bilateral salpingo-oophorectomy and possibly long-term endocrine therapy to improve treatment outcomes. This treatment can lead to premature surgical menopause, which is a major risk factor for sexual difficulties.”

4.     In 5. Assessing Sexual Health in Breast Cancer Survivors

A.     Is there an objective and specific tool for assessing sexual health? If so, it will be need to introduce them in this part.

Author Response

I suggest performing a careful revision to correct some typos and revise grammar.

Authors’ response: Our professional editor and proofreaders read the article and were unable to find any typographical and grammatical errors. If the errors are pointed out, we will be happy to correct them. We did correct misnumbering of headings in section 6 beginning on page 13 with “Pelvic Health Physical Therapy.”

2A. When referring to epidemiology on breast cancer, try to use the latest (2022) data as much as possible.

Authors’ response: These data are from the SEER 2022 cancer statistics; we replaced the original reference 1 with this updated reference from SEER.

2B.  Authors said that the subject of this article is focused on cis-gender women. If so, I recommend to briefly reduce the cisgender men, transgender and gender-diverse contents mentioned in the introduction.

Authors’ response: We agree and have changed this as recommended, by deleting nonrelevant information in the first paragraph and in section 7.

3A. Add a reference to the contents “Occasionally, nerve injury can cause hyperesthesia or dysesthesia of the chest wall, further disrupting potentially pleasurable sexual sensations.”

Authors’ response: We agree and have added a reference (Gong et al) as recommended on page 5.

3B. Add a reference to “This treatment can lead to premature surgical menopause, which is a major risk factor for sexual difficulties.”

Authors’ response: Thank you for this suggestion; we added an additional reference on page 5 (Faubion et al).

Round 2

Reviewer 1 Report

Dear authors,

thank you for the revised version of the submission which adresses my previously mentioned remarks. Reading your current version of the script I am still interested in your search strategy as in pubmed the search for 'anorgasmia,
assessment, breast cancer survivor, cancer treatment, libido, sexual dys-
function, sexual function, sexual health, sexuality, and treatment.' results in 0 results, but also 'anorgasmia breast cancer' results in 0 results. So may I recommend adapting the RAMESES or PRISMA guidelines. This is what I initially ment with my remark, not C&P the paragraph to a different position. 

Thank you for adding a paragraph on imunotherapies. Interestingly you are not mentioning the long term/post treatment side effects like colitis or pulmonary diseases. Which 'naturally' caused have been studied regarding their sexual health effects (PMID: 32736459, PMID: 33031691). Here the use of a validated screening tool helps to compare different subgroups/backgrounds. This is a missing opportunity to foresee possible future problems and as you introduce two screening tools (without references though) point out the benefit/downside of these tools.

On the clinical side you have deleted the contraindicated medication. Still - like in 6.3 '...have not been WELL studied...' in some places you are not providing references for your statements. (i.e. why not well? what study design? screening tool?)

But more importaintly as a clinician/HCP I expect at the end a clear statement which treatment is the most likely to improve the sexual health of my patients. Again here the known improvement of the screening tools could provide a ranking for the decision making.

If paragraph 7 is to be kept, you should provide your own data on this issue in order to generate the first data. But from my viewpoint this is better used as a starting point for a completely seperate article describing the different risk modalities (i.e. HRT) and background risks (male BC vs trans)...

Author Response

RE: Manuscript: Sexual Health After a Breast Cancer Diagnosis: Addressing a Forgotten Aspect of Survivorship

To:

Mirjana Njegovan

Publishing Manager, MDPI Belgrade

It is our pleasure to hear from you that our manuscript may be accepted for publication with major revisions. We are appreciative of the insightful comments from the reviewers and believe their suggestions have led to an improved manuscript. Below is our reply to the reviewer comments:

Reviewer recommendation:

Thank you for the revised version of the submission which addresses my previously mentioned remarks. Reading your current version of the script I am still interested in your search strategy as in PubMed the search for 'anorgasmia, assessment, breast cancer survivor, cancer treatment, libido, sexual dysfunction, sexual function, sexual health, sexuality, and treatment.' results in 0 results, but also 'anorgasmia breast cancer' results in 0 results. So may I recommend adapting the RAMESES or PRISMA guidelines. This is what I initially meant with my remark, not C&P the paragraph to a different position. 

Author response:

We intended to present a comprehensive review of the topic “sexual health in breast cancer survivors” focusing on how the different treatments for breast cancer may influence sexuality and to provide an overall comprehensive plan for managing sexual dysfunction in breast cancer survivors. This is not a systematic review following the PRISMA guidelines, and we chose to do our literature search in PubMed, Google Scholar, and Scopus for peer-reviewed, English-language publications published from January 2010 to March 2022 using relevant search items. Our initial search was to identify literature pertinent to sexual function assessment in breast cancer survivors to gauge the magnitude of their sexual problems. A total of 900 articles were selected and reviewed, and articles not pertinent to breast cancer were deselected. Each of the writing team members focused on various aspects of the breast cancer review.

Our initial individual search terms included anorgasmia, assessment, sexual, libido, breast cancer survivor, mastectomy, lumpectomy, tamoxifen, aromatase inhibitor, sexual dysfunction, sexual function, sexual health, sexuality, treatment, pelvic physical therapy, and genitourinary syndrome of menopause. Additional individual search items included flibanserin, bremelanotide, bupropion, vaginal estrogen, and vaginal DHEA. Combination search terms were pelvic physical therapy AND dyspareunia, pelvic floor physical therapy AND dyspareunia, pelvic floor physical therapy AND dyspareunia AND cancer, and immunotherapy AND breast cancer. We also used the bibliography listed in some of the articles we found to obtain other relevant articles. We extracted and analyzed the data, and summarized this information in an easy-to-read format. In the end, we referenced approximately 160 articles for our manuscript. In the Introduction, we added a methods section, as highlighted, where we described the additional search terms. We hope this clarifies our search methods.

Reviewer recommendation:

Thank you for adding a paragraph on immunotherapies. Interestingly you are not mentioning the long term/post treatment side effects like colitis or pulmonary diseases. Which 'naturally' caused have been studied regarding their sexual health effects (PMID: 32736459, PMID: 33031691). Here the use of a validated screening tool helps to compare different subgroups/backgrounds. This is a missing opportunity to foresee possible future problems and as you introduce two screening tools (without references though) point out the benefit/downside of these tools.

Author response:

Thank you for providing the references, and we have reviewed them in detail. We agree that there are data regarding immunotherapy-caused adverse sexual health effects, such as colitis, pulmonary problems, and fatigue. There are no existing studies, to our knowledge, regarding sexual health and immunotherapy in breast cancer survivors. We revised section 4.5 to note the various adverse effects and their potential impact on sexual health. We also added the need for practitioners involved in caring for breast cancer survivors to have a heightened awareness of these adverse effects, as well as the need for additional research.

Reviewer recommendation:

On the clinical side you have deleted the contraindicated medication. Still - like in 6.3 '...have not been WELL studied...' in some places you are not providing references for your statements. (i.e. why not well? what study design? screening tool?)

Author response:

We have provided additional references for the Female Sexual Function Index and the Decreased Sexual Desire Screener. A full explanation of past study limitations is beyond the scope of the current manuscript.

Reviewer recommendation:

But more importantly, as a clinician/HCP I expect at the end a clear statement which treatment is the most likely to improve the sexual health of my patients. Again, here the known improvement of the screening tools could provide a ranking for the decision making.

Author response:

We understand your concern about providing an overview and recommendations to practitioners. Treatment of sexual difficulties in a breast cancer survivor, however, focuses on a comprehensive approach to the patient with a multidisciplinary team. There is no one drug fits all or best medication, and in our practice, many patients improve without any specific medication management. We apologize for not making this clearer in our initial manuscript. We added an additional paragraph in the beginning of section 6 to clearly state this.

Reviewer recommendation:

If paragraph 7 is to be kept, you should provide your own data on this issue in order to generate the first data. But from my viewpoint this is better used as a starting point for a completely separate article describing the different risk modalities (i.e. HRT) and background risks (male BC vs trans)...

Author response:

We appreciate the reviewer’s thoughts regarding paragraph 7. We do not believe it is appropriate to provide our own original data in this format of a review manuscript, particularly given that difficulties in accessing care are well documented for the LGBTQ+ cancer patient population (as cited in our manuscript).

Women with breast cancer are sexual and gender minorities (in addition to those with cisgender and heterosexual identities), and we have edited the paragraph in section 7 to better streamline and highlight this important point. Every major cancer treatment organization has called for greater representation of LGBTQ+ populations in oncology research and clinical literature. As such, we feel strongly about retaining this information in the manuscript and hope our recent edits provide clarity.

We appreciate your consideration of this manuscript revision for publication in the special edition of Journal of Clinical Medicine. All authors have contributed to the manuscript revisions and have approved the final submission. If you need any additional information or have any questions regarding our submission, please feel free to contact us. Thank you.

Sincerely,

Suneela Vegunta MD, NCMP, FACP
